# Genomic Hatchery Introgression in Brown Trout (*Salmo trutta* L.): Development of a Diagnostic SNP Panel for Monitoring the Impacted Mediterranean Rivers

**DOI:** 10.3390/genes13020255

**Published:** 2022-01-28

**Authors:** Adrián Casanova, Sandra Heras, Alba Abras, María Inés Roldán, Carmen Bouza, Manuel Vera, José Luis García-Marín, Paulino Martínez

**Affiliations:** 1Laboratory of Genetics Ichthyology, University of Girona (UdG), ES-17071 Girona, Spain; adrian.casanova@udg.edu (A.C.); sandra.heras@udg.edu (S.H.); alba.abras@udg.edu (A.A.); marina.roldan@udg.edu (M.I.R.); joseluis.garcia@udg.edu (J.L.G.-M.); 2Department of Zoology, Genetics and Physical Anthropology, Faculty of Veterinary, University of Santiago de Compostela (USC), ES-27002 Lugo, Spain; mcarmen.bouza@usc.es (C.B.); paulino.martinez@usc.es (P.M.)

**Keywords:** brown trout, hatchery introgression, 2b-RADseq, reference genome approach, stacks 2, SNP genotyping, MassARRAY® System

## Abstract

Brown trout (*Salmo trutta* L.) populations have been restocked during recent decades to satisfy angling demand and counterbalance the decline of wild populations. Millions of fertile brown trout individuals were released into Mediterranean and Atlantic rivers from hatcheries with homogeneous central European stocks. Consequently, many native gene pools have become endangered by introgressive hybridization with those hatchery stocks. Different genetic tools have been used to identify and evaluate the degree of introgression starting from pure native and restocking reference populations (e.g., *LDH-C** locus, microsatellites). However, due to the high genetic structuring of brown trout, the definition of the "native pool" is hard to achieve. Additionally, although the *LDH-C** locus is useful for determining the introgression degree at the population level, its consistency at individual level is far from being accurate, especially after several generations were since releases. Accordingly, the development of a more powerful and cost-effective tool is essential for an appropriate monitoring to recover brown-trout-native gene pools. Here, we used the 2b restriction site-associated DNA sequencing (2b-RADseq) and Stacks 2 with a reference genome to identify single-nucleotide polymorphisms (SNPs) diagnostic for hatchery-native fish discrimination in the Atlantic and Mediterranean drainages of the Iberian Peninsula. A final set of 20 SNPs was validated in a MassARRAY® System genotyping by contrasting data with the whole SNP dataset using samples with different degree of introgression from those previously recorded. Heterogeneous introgression impact was confirmed among and within river basins, and was the highest in the Mediterranean Slope. The SNP tool reported here should be assessed in a broader sample scenario in Southern Europe considering its potential for monitoring recovery plans.

## 1. Introduction

Brown trout (*S. trutta*) is a cold-water fish, living in well-oxygenated swift streams, brooks, rivers, and lakes. After the smoltification process, anadromous forms migrate to the sea to feed (i.e., sea trout), foraging close to the coast, not far from the mouth of natal rivers [1] and return to freshwater environments only to spawn. Brown trout’s natural distribution range mainly encompasses Europe as well as Western Asia and North Africa (Atlas Mountains) [2]. Brown trout is considered a genetically and geographically highly structured species [3,4], with the highest differentiation observed among resident brown trout populations [5].

The species has ecological, recreational, and economic importance. Because brown trout is an essential part of ecosystems, it is considered an umbrella [6] or flagship species [7]. Brown trout is categorized as Least Concern (LC) by the International Union for Conservation of Nature (IUCN) Red List of Threatened Species [8]. However, this categorization can change at regional scale (e.g., Vulnerable in Spain [9]). Wild populations are threatened from river fragmentation [10,11], the degradation of spawning habitats [12], water pollution [13,14,15,16], global warming [17], overfishing due to its recreational fishing value [18] and hatchery introgression [19,20,21,22,23]. Due to its recreational fishing interest, brown trout has been introduced into countries outside of its natural range since the 19th century [24,25,26].

Three different scenarios related to releases of non-indigenous individuals can be depicted (revised by [27]: (1) ‘introduction’ when the release is made in a region where the species has never been present; (2) ‘reintroduction’ when the release is carried out in a region where the species was previously present but is currently extinct; and (3) ‘restocking’ when individuals are released into a region occupied by conspecific native populations. The release of hatchery trout has been a common practice from the 20th century to the present day, in order to meet fishing demands and reverse the depletion of wild brown trout populations [28,29,30,31,32,33]. However, restocking can endanger native populations by disrupting local adaptations due to the introduction of maladapted specimens (e.g., [32,34,35]). Most of the hatchery brown trout used for restocking derived from a Central European stock pertaining to the Atlantic lineage, genetically differentiated from native fish [29,36]. The introgression by restocking practices were evaluated using different molecular markers, such as allozymes or microsatellites. The locus *LDH-C** is particularly relevant for its diagnostic role [19,37,38,39,40,41], being fixed for the *100 allele in native Southern European populations and for the *90 allele in the North Atlantic populations [29,41,42,43]. A different introgression degree was reported throughout Southern European rivers, sometimes resulting in the complete replacement of native gene pools [19,21,23,44] within rivers with unstable hydrological conditions [19,23,39,44].

Although restocking practices with allochthonous stocks have mostly been discarded [36,39,44,45] and replaced by other strategies such as supportive breeding [36], it is necessary to develop more sophisticated and cost-effective tools to assess the introgression of natural populations as a result of restocking with the following purposes: (1) to implement recovery measures in accordance with the degree of introgression (e.g., increasing fishing quotas over highly impacted populations or eradicating naturalized populations of hatchery origin); (2) to avoid the use of breeders from introgressed populations in supportive breeding programs or to find local stocks; and (3) to identify introgressed populations acting as black spots contributing to the spread of foreign genetic variants.

Next-generation sequencing (NGS) techniques enable the achievement of genome-wide data collections in a much faster and cost-effective way [46], fueling the transition to conservation genomics approaches [47,48]. Among the high-throughput single-nucleotide polymorphism (SNP) genotyping techniques, restriction site-associated DNA sequencing (RADseq; [49]) plays a prominent role in this field [50,51], with both model and non-model organisms [52]. With the attainment of genome-wide SNP data and the development of bioinformatic tools for their handling [53], higher resolution approaches can be applied for the assessment of introgressed populations [54]. Recently, the brown trout genome was published [55], increasing the robustness of SNP genotyping from RADseq data.

In this study, the 2b-RADseq genotyping-by-sequencing (GBS) methodology [56] was used to identify a set of SNPs capable of distinguishing a reference hatchery stock and the main native brown trout lineages inhabiting the Iberian Peninsula. Using this set as a starting point, we selected a panel with the highest discriminating SNPs between hatchery and wild locations to be applied in a cost-effective MassARRAY® genotyping (Agena Bioscience®; [57]). The performance of this tool was contrasted against the total genotyping data and used for estimating introgression in a set of samples with information from other molecular markers (locus *LDH-C** and microsatellite genotypes).

## 2. Materials and Methods

### 2.1. Samples

A total of 299 individuals captured from fourteen Iberian locations of *S. trutta*, including the four main mitochondrial DNA (mtDNA) lineages in the Atlantic (Atlantic or AT, Duero or DU) and Mediterranean (Adriatic or AD, Mediterranean or ME) drainages (see [2] for a complete description of the different mtDNA lineages described for the species) were used for the 2b-RAD analysis (Table 1, Figure 1). Temporal replicates were available in two locations from Ter River (NU04-NU14 and TE04-TE14, respectively; see Table 1). Different restocking impacts with the Central European hatchery stock were previously assessed for the analyzed Atlantic [33,58,59,60] and Mediterranean [45] locations. Two representative samples from the Bagà hatchery stock (BA14 and S) used for restocking in Catalonia (Autonomous Community of Spain) were included in our analyses. Despite there being several hatcheries across the Iberian Peninsula, their genetic constitution is quite homogeneous considering the high structuring of the species (G_ST_ = 0.03; [29]). Thus, the Bagà hatchery was considered representative of fish used for restocking in the Iberian Peninsula. For MassARRAY genotyping, 81 reference individuals from hatchery and wild basins, previously genotyped using Stacks from the 2b-RADseq dataset, were used for genotype validation along with 241 individuals to compare the performance of our molecular tool with previous methodologies (Table 1).

### 2.2. Library Preparation and Reads Processing

DNA extraction and 2b-RAD libraries preparation followed the protocol described by Wang et al. (2012) [56] with slight modifications. AlfI IIB restriction enzyme (RE) was used for digestion of genomic DNA in 36 bp length fragments to construct the libraries to be sequenced on a NextSeq500 Illumina sequencing platform following a 50 bp single-end chemistry. The number of recognition sites in *S. trutta* genome was obtained using the last version of ExtractSites.pl script (https://github.com/Eli-Meyer/2bRAD_utilities, accessed on 15 June 2020) taking the brown trout genome as reference ([55]; NCBI GenBank assembly accession: GCA_901001165.1). After raw data demultiplexing, several filtering criteria were applied: (1) all reads were trimmed to the expected length (i.e., 36 nucleotides), (2) filtered by the RE recognition site presence using home Perl scripts, and (3) cleaned with process_radtags (module belonging to Stacks 2.41; [64,65,66]) to remove reads with at least one uncalled nucleotide or nine or more consecutive nucleotides with average Phred quality scores lower than 30 (base call accuracy of 99.9%; −w 0.25, −s 30, −c, −q). Filtered reads were aligned to the brown trout genome with Bowtie 1.3.0 [67] using the −v alignment mode (v = number of allowed mismatches; −v 3). To limit artifactual SNPs from the tetraploid origin of the salmonids [68,69,70], reads with more than one alignment with the same quality were discarded (—best —strata m 1). Individuals with less than 900,000 aligned reads (Q1 aligned reads—IQR) were removed, since individuals with a low number of reads might act as a burden to obtaining a robust SNP panel (see Table 1).

### 2.3. SNP Calling and Genotyping: Defining a SNP Panel for All Locations Studied

Stacks 2 input alignments were oriented in the same way using a home Perl script and sorted by chromosome using Samtools 1.10 [71]. The Stacks pipeline for further analyses consisted in two modules: (1) gstacks, which genotypes the SNPs identified per locus in each individual; and (2) populations, which obtains a variety of standard output formats for further population genetics analyses (e.g., Genepop format). Different Genepop file SNP subsets were obtained with Genepopedit 1.0 R package [72] and converted into different file formats using PGDSpider 2.1.1.5 software [73]. The initial panel of obtained SNPs was analyzed to detect the same SNPs from overlapping RAD-loci, caused by nearby AlfI targets. Moreover, error genotyping rate was calculated for this initial panel of SNPs using the script “ErrorCount.sh” included in dDocent pipeline [74]. Next, the raw SNP panel was filtered to retain a consistent set of markers and alleles represented across the individuals genotyped according to the following criteria: (1) ≥6 reads coverage per locus and individual, (2) least-frequent allele score ≥ 3 alleles in the whole sample (minimum allele count, MAC ≥ 3), (3) genotyped in at least 60% of the individuals in each of the seventeen samples, (4) conformance to Hardy–Weinberg (HW) expectations using a combined *p* > 0.05 across the samples analyzed (Fisher’s method [75]), (5) ≤3 SNPs per RAD-locus, (6), the most polymorphic SNPs were retained when several SNPs occurred in a single RAD locus to avoid redundant information, and (7) the SNPs were homologous to a single position in the main scaffolds of the brown trout genome assembly correspondent to the *n* = 40 chromosomes of the species [76] (see Appendix A for information about code used in this section).

### 2.4. Detection of SNPs Variation between Hatchery and Native Fish

The Bayesian clustering method implemented in STRUCTURE 2.3.4 [77] using R package ParallelStructure 1.0 [78] was applied to identify individuals with hatchery ancestry (*qH*, admixture coefficient) in wild locations using the BA14 hatchery as reference. This made it possible to look for diagnostic loci differentiating hatchery and native individuals from the Iberian Peninsula. A total of 244 individuals coming from 13 wild locations from both the Atlantic and Mediterranean slopes were analyzed. First, using all specimens (N = 283; POPINFO = 0), we determined the minimal K value where all hatchery specimens were clustered together and separated from the wild samples. The lowest *qH* detected among BA14 individuals was used as a threshold to identify pure fish for further analyses. Then, from the STRUCTURE analyses of wild locations (see below), all individuals were classified into three categories: (1) hatchery individuals (*qH* > 0.95), admixed (*qH* between 0.05 and 0.95), and (3) native individuals (*qH* < 0.05). Next, for each wild location, an admixture model with two origins was performed (hatchery vs. native; K = 2), where hatchery individuals were forced to be non-admixed (i.e., POPFLAG = 1), following an incomplete, frequently used baseline method [79]. Due to the high genetic differentiation reported between Iberian native populations and the hatchery stocks, a model of independent allele frequencies was used (see [79]). Ten independent replicate runs were performed to limit the influence of stochasticity and increase the precision of estimations following recommendations [80]. A burn-in of 100,000 iterations followed by 200,000 Markov Chain Monte-Carlo steps (MCMC) were applied for these analyses. SNPs from the whole dataset that showed a single copy of the least-frequent allele (singletons) when using hatchery references and wild samples were removed for analyses as recommended by Linck and Battey (2019) [81].

Conformance to HW expectations was checked at each wild location using an exact probability test as implemented in the R package Genepop 1.1.7 (based on the Genepop 4.7.5; [82]). Tests were performed considering all individuals and after removing specimens of hatchery ancestry (*qH* > 0.05). SNPs were classified according to genetic differentiation (F_ST_, [83]) between the hatchery stock (BA14 and S specimens) and a pool of all native fish (*qH* < 0.05). Later, for each location, three additional STRUCTURE analyses were run (for K = 2) with three different subsets of physical, unlinked SNPs: F_ST_ > 0.95, F_ST_ > 0.99 and F_ST_ = 1.00 (i.e., fully diagnostic SNPs fixed for alternative allelic variants in wild and hatchery fish) and results were compared with those obtained using the whole SNP dataset. The matching ratio of fish classified as native, admixed and hatchery with each subset of SNPs regarding the whole SNP dataset information was used to establish their performance for the classification of fish. Annotation of the SNPs selected was performed with the Ensembl Variant Effect Predictor 104 software (VEP; [84]).

### 2.5. Multiplex MassARRAY Design and Implementation

To validate 2b-RAD genotyping, we used the MassARRAY technology on the 81 control individuals (16 hatchery + 65 native) derived from different locations of the Mediterranean and Atlantic slopes. In brief, the protocol consists of a two-step reaction: The first involves the amplification by polymerase chain reaction (PCR) of an amplicon, including the selected SNP, and the second involves a mini-sequencing reaction using an internal primer adjacent to the SNP, which extends the primer with a dideoxinucleotide complementary to the SNP variant [85]. Flanking regions of 100 nucleotides in length of the selected SNPs were obtained from the brown trout reference genome using a home Perl script (201 nucleotides in total length; see Appendix A). Three criteria were used to select the SNPs to be genotyped following this methodology: (1) no genetic variation in the flanking regions, as identified in the raw panel of Stacks 2.41, to avoid interference with primer annealing [85]; (2) flanking regions displaying a unique BLAST-alignment against the reference genome [86]; (3) unlinked SNPs in order to get a panel as informative as possible. Finally, for a cost-effective genotyping, 20 diagnostic SNPs were selected. MassARRAY genotyping was conducted at the UCIM-University of Valencia Genomics Platform to validate 2b-RAD genotyping.

Secondly, the 241 individuals from eleven locations (Table 1, Figure 1), with previous information available on hatchery ancestry from five microsatellite loci and the *LDH-C** locus were used to check the performance of the new tool [45]. For this, we chose samples from rivers draining to the Mediterranean slope because of their higher impact on restocking within Iberian Peninsula [19,44]. Hatchery ancestry (*qH*) at individuals and locations was estimated in the eleven locations by using previous hatchery and native controls as references and keeping the same above-described parameters for STRUCTURE software. Correlations between the different estimators of hatchery ancestry using different molecular makers (i.e., SNPs from the panel, microsatellites and *LDH-C** locus) were performed with R.

## 3. Results

### 3.1. Identification of SNPs to Assess Hatchery Ancestry in Brown Trout from Iberian Peninsula

The number of AlfI RAD-loci identified in the brown trout genome was 557,761, representing ~0.8% of the total genome assembly. Sixteen individuals with less than 0.9 M aligned reads were removed (see Table 1; Materials and Methods section). A total of 361,129 RAD-loci were built by Stacks 2.41, comprising 606,037,805 aligned reads and representing an average 9.0× (±2.6×) coverage per locus and individual. After Stacks 2.41 parsing, 191,406 SNPs were called. Among these, 1150 SNPs (0.6%) were shared by overlapping RAD-loci, being discarded for further analyses. The error genotyping rate ranged from 0.050 to 0.120 using the “ErrorCount.sh” script. After applying all quality and population filters, the final dataset was composed of 24,830 nuclear SNPs mapped on the 40 assembled chromosomes of the brown trout genome.

Using the whole SNP dataset, a total of 34 individuals of hatchery ancestry with the aforementioned threshold were identified in the wild locations studied. Of all individuals captured at VI, CH, AG1, OM, P2 and P3, TE04 and TE14 were native, while 14 out of the 18 individuals from QB14 showed different degrees of hatchery ancestry (*qH* > 0.05). Accordingly, this location was not used to identify diagnostic allele variants between native and hatchery fish. One individual from FE, three from LE, two from BL (the same reported by Martínez et al. 2007 [59]), five from CE, five from NU04 and four from NU14 were also identified as admixed and removed to look for a differentiation between pure native and hatchery fish. Nevertheless, all introgressed locations were in HW equilibrium (*p* = 1.00), suggesting random mating and the admixture of hatchery variants into the wild after several generations since the start of releases (data not shown). All of the SNPs used were ranked by F_ST_ (hatchery vs. native; (median F_ST_ = 0.091) to select those with the highest diagnostic power (F_ST_ > 0.95) between these two groups. The number of SNPs with F_ST_ > 0.95, F_ST_ > 0.99 and F_ST_ = 1.00 were 214, 38 and nine, respectively (Appendix A). These 214 SNPs were mapped in 37 chromosomes of the brown trout genome. Among them, 47.4% were mapped in intergenic regions (IGRs), 46.0% within introns and 6.5% within exons (Appendix A), mostly being non-synonymous variants. The performance of these three SNP panels to assess hatchery ancestry was similar when compared to the whole SNP dataset (Figure 2). In 10 out of 15 samples (VI, FE, CH, AG1, OM, P2, P3, NU04, TE04, and TE14), a full agreement was observed for the three SNP panels. Although the best performing panel was that with 214 SNPs, the 38 SNPs panel performed very similarly, and only one individual was classified into a distinct group (NU14-01). The agreement was also high with the nine fully diagnostic SNP panels (F_ST_ = 1.00). Discordances mainly occurred in locations with an average hatchery ancestry (*qH*) > 0.05, especially at the CE location.

Agreement with the whole SNP dataset at CE was slightly lower than 80% with all the three SNP panels (Figure 2). In this location, several individuals showed a *qH* close to 0.05, the threshold established to distinguish between native and admixed individuals. Thus, while five individuals were considered admixed (*qH* between 0.05–0.95) using the whole panel, only one was identified with the 214 and 38 SNP panels, while none were identified with the total of nine diagnostic SNPs panels. Some discordances at other locations were also explained by this fact (e.g., NU14-01). Despite the misclassification of a few individuals, the estimated *qH* for all wild individuals (N = 244) from the different SNP panels was significantly correlated with the whole dataset (*p* for all Spearman’s ρ values < 0.001).

### 3.2. Design and Validation of a Cost-Effective Tool Using Massarray Genotyping

Adjacent genomic regions (±100 bp) to the 214 selected SNPs (F_ST_ > 0.95) were revised to check for the lack of polymorphisms and multiple possible alignments along the genomes to avoid interference with primers annealing for the MassARRAY genotyping technology. A total of 39 SNPs fulfilled this criterion, becoming potential candidates to be included in the MassARRAY analysis. Among them, 20 SNPs located in 18 different brown trout chromosomes, far away when mapping in the same chromosome, were compatible for a single multiplex reaction for MassARRAY genotyping, thus being suitable for a cost-effective evaluation of hatchery ancestry (Appendix A). MassARRAY genotyping mostly matched with 2b-RAD GBS, excluding the SNP TRU08, which showed the same homozygous genotype for nearly all native and hatchery controls, being discarded. Genotyping concordance for the remaining 19 SNPs was 98.60% without considering missing genotypes (Figure 3, Appendix A), which were the major source of differences (2.47%). No genotyping biases were observed between both approaches. Thus, these 19 SNPs were used for a routine MassARRAY genotyping on a larger brown trout sample to check for their performance in comparison with previous microsatellite and *LDH-C** markers.

A total of 241 individuals from 11 wild locations with previous microsatellite and *LDH-C** information were genotyped by MassARRAY for the 19 selected SNPs. One individual (ME16-674) showed a very poor genotyping performance and was discarded for further analyses (see Appendix A; final N = 240). Hence, the SNP genotyping call rate was 98.49% (98.89% when ME16-674 was excluded), with most individuals showing no missing data (79.67% and 80.00% with and without ME16-674, respectively), 18.67% (18.75% without ME16-674) showing only one missing call, and only 1.66% (1.25% without ME16-674) with two or more missing genotypes (Appendix A).

The hatchery impact in the 11 new locations estimated as the average *qH* per locus and individual with these 19 SNPs was heterogeneous, ranging from 0.024 in ME16 to 0.980 in RQS18, the latter likely being a naturalized hatchery location (Table 2). All locations showed some degree of introgression derived from past hatchery releases (Appendix A), and the results were fairly coincident with previous estimates from *LDH-C***90* allele frequency and microsatellite loci at the location level. The highest correlation was between SNPs and *LDH-C***90* estimations (Spearman’s ρ = 0.963; *p* < 0.001), being much lower with microsatellites (Spearman’s ρ = 0.500; *p* > 0.05). The lower correlation with microsatellites was mainly due to the ME16 and GRE17 locations (Table 2). When both locations were removed, the correlation with microsatellites greatly increased (Spearman’s ρ = 0.933; *p* < 0.001). At the individual level, the correlation with estimates was worse between SNPs and *LDH-C***90,* but significant (Spearman’s ρ = 0.722, *p* < 0.001). The correlation between SNPs and microsatellites with all locations was lower (Spearman’s ρ = 0.448, *p* < 0.001; N = 240) than when the ME16 and GRE17 locations were removed, as outlined above (Spearman’s ρ = 0.801, *p* < 0.001; N = 187).

## 4. Discussion

As a result of the evolutionary processes associated with the Pleistocene glaciations, the brown trout is most likely the vertebrate species with the highest population genetic structuring throughout Europe; several of its long-time diverging lineages were previously described [2,4,87,88]. The Iberian rivers harbor one of the richest and most diverse genetic constitutions of brown trout, and previous analyses from sequence variation in the mtDNA allowed for the detection of four native lineages: Adriatic (AD), Mediterranean (ME), Atlantic (AT), and the endemic Duero (DU) [62,63,88,89,90,91]. The AD and ME lineages are widely distributed throughout all Southern European rivers [92], and the native Iberian populations of the AT lineage represent a long-time divergent branch from the older AT clade distributed among Central and North European populations [90]. Substantial genetic divergence was also observed at nuclear loci [4,58,59,60,79,93]. In our study, using a 2b-RAD method for the first time for high-throughput SNP genotyping, 24,830 nuclear SNPs, distributed among the 40 assembled chromosomes of the brown trout genome, were retained after applying all quality and population filters. The SNP density obtained was within the broad range of other brown trout genome-wide studies, from ~3 K [94,95] to ~100 K [52,96]. Further, this study is a pioneering investigation, using the brown trout reference genome for genotyping (see [97]) due to its recent availability (NCBI June 2019; [55]).

To counterbalance brown trout population declines, Mediterranean river basins were extensively restocked during the 20th and at the beginning of the 21st centuries, using non-indigenous hatchery stocks from North Atlantic basins [36,44,98,99]. All of these stocks share a homogeneous genetic background [29,36]. The genetic impact of restocking in Mediterranean countries is frequently evaluated using the nuclear diagnostic marker *LDH-C**. The use of a high number of SNPs distributed across the whole brown trout genome, and the availability of a reference hatchery sample from Bagà to estimate the genetic incidence of restocking across the Iberian populations in the Atlantic and Mediterranean slopes, rendered similar values at population level. Our results corroborated a higher impact in Mediterranean-flowing rivers, as previously reported [19], and a distinct impact between locations separated by just a few kilometers in these basins (e.g., NU14 and QB14, see [45]). Despite the high correlation obtained with *LDH-C** results at population level, the use of *LDH-C** genotypes were less accurate to typify individuals in introgressed populations since, for instance, 50% of the F2 from a native x hatchery cross are expected to be homozygotes for the native *LDH-C*100* or **90* alleles, despite their 50% average hatchery genomic background. Furthermore, with only this locus, some approaches (e.g., the characterization of admixed individuals for hybrid classification) were impossible to achieve. The SNPs detected in this research allowed for a more accurate classification of individuals by hatchery ancestry using the probability of membership assignment with STRUCTURE, as previously reported [100,101]. Our conservative *qH* threshold was more restrictive than those previously used for the species [79] and other fishes (e.g., *Scophthalmus maximus*; [101]) to identify individuals with pure native or hatchery ancestry.

Notably, the 214 SNPs identified here showed an almost full divergence (F_ST_ > 0.95) between hatchery and native fish, irrespective of their lineage, as previously described for the *LDH-C** locus. The efficiency of these markers, as well of their two subsets, were composed by SNPs with a progressively higher diagnostic power (from F_ST_ > 0.99 (38 SNPs) to F_ST_ = 1.00 (9 SNPs)) to relatively determine native, hatchery and admixed fish with conclusions from the whole SNPs dataset. The discrepancies were related to individuals with a hatchery impact (*qH*) close to the threshold used in the classification, which were very frequent in CE location. Furthermore, because introgression is not homogeneous across the genome [98], a portion of genome covered by these SNPs may not follow the global trend and produce a particular *qH* estimate affecting individuals’ classification.

The genotyping performance of the developed MassARRAY tool was very high, with only one fish failing almost all markers (ME16-674, Appendix A), probably due to the low quality of its available Chelex DNA extraction [102]. All but one of the 20 selected SNPs performed very well, resulting in genotypes already detected by the 2b-RAD methodology. Substantial mismatches from previous 2b-RAD genotypes were identified at the SNP TRU08 which was discarded for future routine analyses. This phenomenon was previously reported [103], where after MassARRAY validation, 10% of SNPs detected by ddRADseq became homozygous, plausibly due to low coverage. There are several processes that could explain the failure at the SNP TRU08. For instance, the primers designed for TRU08 could amplify some other similar but monomorphic regions that were not detected in the BLAST analysis. This could happen due to upstream or downstream SNPs not detected by 2b-RADseq in the target region, and there may also be some similar regions of the genome that are not sufficiently present in brown trout assembly. Another explanation could be that this SNP was a bioinformatic artefact, the consequence of over-merged loci (i.e., paralogous), being the diagnostic polymorphism in another region of the genome. Despite brown trout being a species of tetraploid origin [70], the coverage obtained with this marker from 2b-RADseq (12.46× ± 4.63× across 248 genotyped individuals) make the latter explanation unlikely. Nonetheless, TRU08 could be replaced by another diagnostic SNPs from the 2b-RADseq catalog (Appendix A) or even by the SNP responsible for the distinct electrophoretic pattern between *LDH-C***100* and **90* alleles [43].

In any case, the MassARRAY tool using 19 SNPs successfully determined the hatchery impact across 11 Mediterranean locations (N = 240 individuals) without previous 2b-RAD information. The MassARRAY approach is a more complete and cost-effective way to determine hatchery impact and identify native brown trout in Mediterranean rivers, compared to using 2b-RADseq or other RADseq methods, which are economically unfeasible for the monitoring of wild populations. In our laboratories, without taking into account DNA extraction, the current price of RE digestion for *LDH-C** genotyping using the methodology developed by McMeel et al. (2001) [43] would be around EUR 7/individual to test just one SNP, while using the developed tool would be around EUR 5/individual for genotyping 20 SNPs, reducing drastically the cost per SNP. Other SNP tools were successfully applied for the identification of hybrids in aquatic organisms [104,105,106,107]. In brown trout, the detection of diagnostic native alleles may support the successful management practices addressed to identify black spots from past releases leading to almost naturalized populations (such as RQS18 in the present study). Moreover, these variants could also identify remaining native specimens to be used in supportive breeding or to find regional stocks to replace the foreign ones and adjust fishing status and quotas according to the hatchery impact in the management regulations (e.g., ARP/260/2021, DOGC resolution [108]).

## 5. Conclusions

The present results support the usefulness of the SNP MassARRAY tool developed here for the estimation of stocking from a large sample of individuals covering the Iberian Peninsula. However, because AD and ME trout lineages are distributed throughout the Mediterranean basin, our tool could also be useful for studying other Mediterranean trout populations outside of Iberian rivers. Furthermore, in case additional markers are needed, there is a larger diagnostic SNP catalog that can be included in this tool or in a brown trout, high-density array. In addition, this study highlights the usefulness of RAD-seq strategies to provide such a volume of genomic data that many objectives can be covered.

## Figures and Tables

**Figure 1 genes-13-00255-f001:**
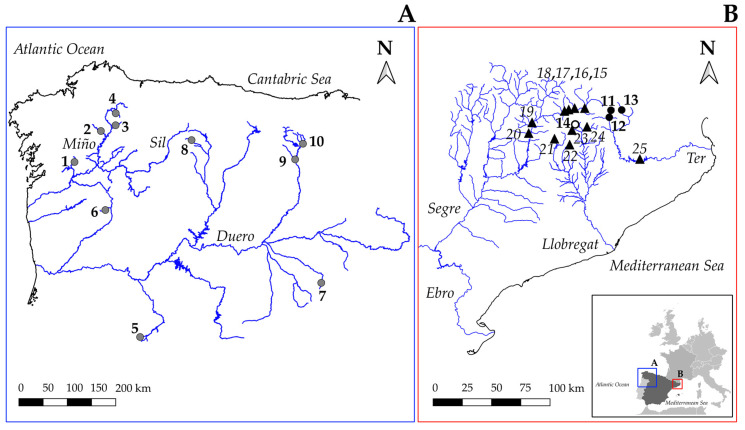
Studied sampling locations from Iberian Peninsula in the Atlantic (**A**) and Mediterranean (**B**) drainages. Numerical codes are shown in Table 1. The names of the main rivers are also included. Circles in 1A and 1B indicate samples used for 2b-RADseq analysis. Triangles in 1B indicate samples used for MassARRAY genotyping. Grey and black colors indicate locations belonging to the Atlantic and Mediterranean drainages, respectively. The empty circle (no. 14) indicates the location of Bagà hatchery.

**Figure 2 genes-13-00255-f002:**
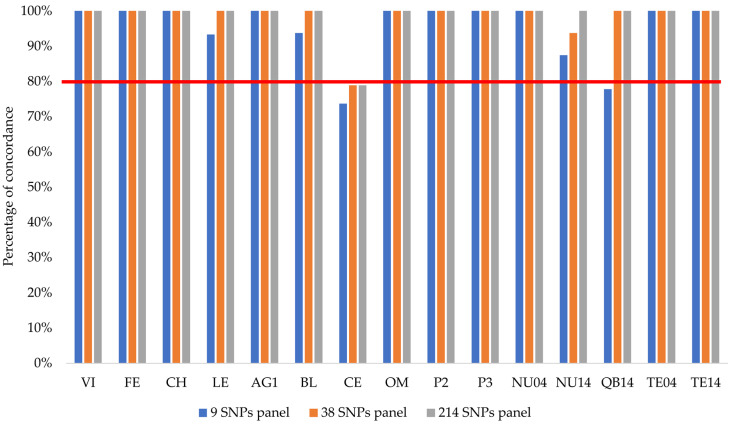
Percentage of concordant classification of hatchery ancestry individuals with the three SNP panels compared with the whole SNP dataset. The red line shows the 80% of concordance for all wild samples studied.

**Figure 3 genes-13-00255-f003:**
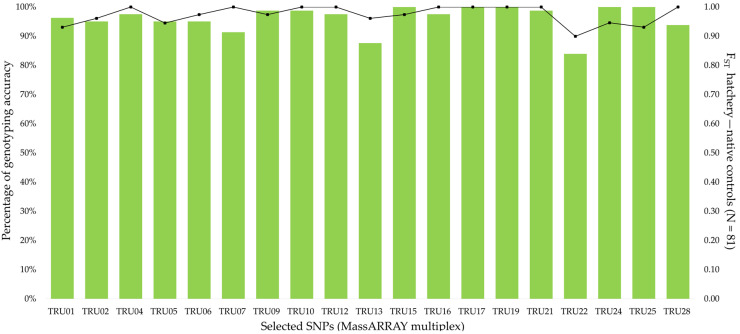
Concordance between 2b-RAD and MassARRAY genotyping for the 19 SNPs selected. Bars represent the percentage of matching genotypes between MassARRAY and 2b-RAD data. The dark line at the top corresponds to the F_ST_ between hatchery and native controls.

**Table 1 genes-13-00255-t001:** Brown trout (*S. trutta*) locations analyzed in this study. Numbers correspond to those used in Figure 1 to identify sampling locations. N: Number of individuals; the final number used for bioinformatic analysis in brackets. Native mtDNA lineage corresponding to sampling locations is indicated [36,60,61,62,63]. NA: Not Available.

Locations	Year	Code	N	Native mtDNA Lineage
**Samples for 2b-RADseq analysis**				
Miño-Sil River basin			*59 (56)*	
1. Viñao River	2003	VI	16 (15)	AT
2. Ferreira River	2003	FE	14 (13)	AT/DU
3. Chamoso River	2003	CH	13 (13)	AT/DU
4. Lea River	2003	LE	16 (15)	DU
Duero River basin			*119* (*106*)	
5. Águeda River	2002	AG1	20 (16)	AT
6. Porto do Rei Búbal River	2002	BL	19 (16)	AT
7. Cega River	2002	CE	20 (19)	AT/DU
8. Omaña River	2002	OM	20 (20)	DU
9. Pisuerga River 2	2002	P2	20 (18)	DU
10. Pisuerga River 3	2002	P3	20 (17)	DU
Ter River basin			*82* (*82*)	
11. Núria River	2004	NU04	16 (16)	AD/ME
11. Núria River	2014	NU14	16 (16)	AD/ME
12. Queralbs, in Freser River	2014	QB14	18 (18)	AD/ME
13. Ter River	2004	TE04	14 (14)	AD/ME
13. Ter River	2014	TE14	18 (18)	AD/ME
Hatchery			*39* (*39*)	
14. Hatchery release individuals	2014	BA14	19 (19)	AT
14. Hatchery spawners	2002	S	20 (20)	AT
**Samples for MassARRAY**				
Ebro River basin			*130*	
15. Segre, Queixans	2016	QU16	30	AD/ME
16. Segre, Meranges	2016	ME16	24	ME
17. Segre, Prullans	2016	PR16	30	ME
18. Segre, Martinet	2017	MA17	33	AD/ME
19. Segre, Els Hostalets de Tost	2018	TS18	4	ME
20. Segre, Organyà	2018	OR18	9	ME
Llobregat River basin			*95*	
21. Cardener River	2018	CA18	17	ME
22. Aiguadora; Bancells Mill)	2017	CT17	14	ME
23. Gressolet	2017	GRE17	30	ME
24. Riutort	2017	RT17	34	ME
Ter River basin			*16*	
25. Querós Creek	2018	RQS18	16	NA

**Table 2 genes-13-00255-t002:** Inter-marker comparisons of hatchery impact estimations. Data from microsatellites and *LDH-C***90* allele are derived from [63] and unpublished data. For more information about microsatellites and *LDH-C** methodology see [45].

Location Codes	*qH* (19 SNPs)	*qH* (5 Microsatellites)	*LDH-C*90*
QU16	0.056	0.027	0.050
ME16	0.024	0.381	0.022
PR16	0.241	0.074	0.217
MA17	0.210	0.071	0.197
TS18	0.496	0.159	0.250
OR18	0.439	0.293	0.556
CA18	0.147	0.029	0.059
CT17	0.120	0.058	0.143
GRE17	0.570	0.097	0.667
RT17	0.545	0.210	0.397
RQS18	0.980	0.400	0.969

## Data Availability

Biological and genomic information (including access to NCBI sequence read archive (SRA) database) for the samples used in the current study is available in the NCBI Bioproject database (ID: PRJNA784924; https://www.ncbi.nlm.nih.gov/bioproject/PRJNA784924/ accessed on 15 June 2020). Brown trout reference genome sequence was downloaded from the NCBI genome assembly website (https://www.ncbi.nlm.nih.gov/assembly/GCF_901001165.1/ accessed on 15 June 2020).

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
