# Peer review of "Genomic Hatchery Introgression in Brown Trout (Salmo trutta L.): Development of a Diagnostic SNP Panel for Monitoring the Impacted Mediterranean Rivers"

_genes, 2022, doi:10.3390/genes13020255_

Round 1
Reviewer 1 Report
The samples used in the study are not enough to export reliable population genetic parameters. Therefore, the increase of sample size will increase reliability. Therefore, the manuscript could not be published under these circumstances.
Author Response
First of all, we wish to thank the reviewer’s evaluation. In order to provide clarity, we have included the referee's comment in italics and our answer just below in bold. We have also included line numbers in the new version to facilitate the revision task in our reviewer’s answers.
The samples used in the study are not enough to export reliable population genetic parameters. Therefore, the increase of sample size will increase reliability. Therefore, the manuscript could not be published under these circumstances.
We agree with the reviewer that sampling sizes around 20-30 individuals are recommended, basically when a reduced number of molecular markers are used (Hale et al. 2012). However, when a high number of molecular markers are used (> 1,000 SNPs) a reduced number of individuals (4-8) are enough to get confident population genetics estimators (Willing et al. 2012; Nazareno et al. 2017). In the present study, we used 24,830 SNPs in 299 individuals from 14 locations to identify the SNPs useful for genetic introgression. The sampling size (N) in these locations ranged from 13 to 20 (see Table 1). After that, we analyzed 241 individuals from 11 locations and only one (TS18) had a reduced N composed by 4 specimens. It must be also taken into account the fish availability in the different locations, being some of the analyzed point very depleted. Therefore, we think that the total number of individuals analyzed (540) and markers are enough to get reliable estimates.
References:
Hale, M.L.; Burg, T.M.; Steeves, T.E. Sampling for microsatellite-based population genetic studies: 25 to 30 individuals per population is enough to accurately estimate allele frequencies. PLoS ONE 2012, 7, e45170.
Nazareno, A.G.; Bemmels, J.B.; Dick, C.W.; Lohmann L.G. Minimum sample sizes for population genomics: an empirical study from an Amazonian plant species. Mol. Ecol. Resour. 2017, 17, 1136-1147.
Willing, E-M.; Dreyer, C.; van Oosterhout, C. Estimates of genetic differentiation measured by FST do not necessarily require large sample sizes when using many SNP markers. PLoS ONE 2012, 7, e42649.

Reviewer 2 Report
Casanova et al. reports on the development of a genome-enabled diagnostic SNP panel for monitoring genomic hatchery introgression in brown trout in the Mediterranean.
I enjoyed reviewing and studying their manuscript, which offers a fresh perspective on cost-effective genotyping platform and the use of genome-enabled research. I did not find any significant conceptual or methodological flaws, and I only offer a few minor quibbles for the authors to consider, some of which are compulsory.
### Quibbles
- Compulsory revision: I would like the authors to deposit their bioinformatics code for the sections 2.3 and 2.5 in a public repository such as Dryad (https://datadryad.org/stash), figshare (https://figshare.com/) or GitHub (https://github.com/) to facilitate the adoption of their approach to other study systems impacted by aquaculture or restoration practices. Please proved an annotated script(s) including information how the use of your code can be cited or acknowledged.
- Compulsory revision: Given the low coverage (< 10 X), may you please provide the error rates using Jon Puritz’s script found here: https://github.com/jpuritz/dDocent/raw/master/scripts/ErrorCount.sh
- Recommendation: I believe that more clarity and discussion is needed on the treatment of admixed individuals, and the threshold of qH > 0.95 that fails to account for multigenerational hybrid individuals, no? Biologically, I think it may be wise to use HYBRIDDETECTIVE (https://doi.org/10.1111/1755-0998.12704) instead of STRUCTURE to assess the accuracy, efficiency and power to correctly identify hybrid classes/introgression. What do you think?
### Verdict
I congratulate the authors for their outstanding work, and I appreciate their attention to detail in the analytical workflow. I believe many aquaculture genetics practitioners would like to apply the approach reported in the present study in their own study systems. It is important that you give as much detail as you can for the code used in the present study.
Author Response
First of all, we wish to thank the positive reviewer’s evaluation. In order to provide clarity, we have included the referee's comment in italics and our answer just below in bold. We have also included line numbers in the new version to facilitate the revision task in our reviewer’s answers.
Casanova et al. reports on the development of a genome-enabled diagnostic SNP panel for monitoring genomic hatchery introgression in brown trout in the Mediterranean.
I enjoyed reviewing and studying their manuscript, which offers a fresh perspective on cost-effective genotyping platform and the use of genome-enabled research. I did not find any significant conceptual or methodological flaws, and I only offer a few minor quibbles for the authors to consider, some of which are compulsory.
### Quibbles
Compulsory revision: I would like the authors to deposit their bioinformatics code for the sections 2.3 and 2.5 in a public repository such as Dryad (https://datadryad.org/stash), figshare (https://figshare.com/) or GitHub (https://github.com/) to facilitate the adoption of their approach to other study systems impacted by aquaculture or restoration practices. Please proved an annotated script(s) including information how the use of your code can be cited or acknowledged.
Following reviewer’s recommendation, codes for section 2.3 and 2.5 are provided as the Supplementary material Text S1 and GitHub repository (https://github.com/adriancasanovachiclana/Genomic-scripts). The three criteria used in 2.5 section to select the SNPs to be genotyped were carried out using the vcf. file within excel (1 and 3) and online BLAST tool (2). The selection and development of SEQUENOM multiplex reaction was carried out by the University of Valencia service, and we have no information about the selection. Anyway, all the genomic information for each marker can be found in supplementary Table S1.
Compulsory revision: Given the low coverage (< 10 X), may you please provide the error rates using Jon Puritz’s script found here: https://github.com/jpuritz/dDocent/raw/master/scripts/ErrorCount.sh
The error rates (0.050 - 0.120) have been included using the pipeline dDocent (Puritz et al. 2014) applying the “ErrorCount.sh” script (see current lines 160-161 and 250-251), which is useful for RADtags with a coverage ≤ 5X. Below you can find the output for our complete dataset:
#RESULTS FOR COMPLETE PANEL
This script counts the number of potential genotyping errors due to low read depth
It report a low range, based on a 50% binomial probability of observing the second allele in a heterozygote and a high range based on a 25% probability.
Potential genotyping errors from genotypes from only 1 read range from 1431986.0 to 2147979.0
Potential genotyping errors from genotypes from only 2 reads range from 685906.5 to 1543289.625
Potential genotyping errors from genotypes from only 3 reads range from 349736.875 to 1175115.9
Potential genotyping errors from genotypes from only 4 reads range from 181240.875 to 916353.8640000001
Potential genotyping errors from genotypes from only 5 reads range from 93382.46875 to 708212
283 number of individuals and 191406 equals 54167898 total genotypes
Total genotypes not counting missing data 54167898
Total potential error rate is between 0.05062505321417493 and 0.11983020624134244
SCORCHED EARTH SCENARIO
WHAT IF ALL LOW DEPTH HOMOZYGOTE GENOTYPES ARE ERRORS?????
The total SCORCHED EARTH error rate is 0.2638755891912217.
The 24,830 markers included in our final SNP panel had a coverage > 6X (as it is shown on the text, see current lines 163-164). Therefore, the Puritz’s script was not useful. Anyway, we also include the obtained results where you can see that error rate estimated was 0 (data not included on the text)
# RESULTS FOR FINAL PANEL
This script counts the number of potential genotyping errors due to low read depth. It report a low range, based on a 50% binomial probability of observing the second allele in a heterozygote and a high range based on a 25% probability. Potential genotyping errors from genotypes from only 1 read range from 0.0 to 0.0
Potential genotyping errors from genotypes from only 2 reads range from 0.0 to 0.0
Potential genotyping errors from genotypes from only 3 reads range from 0.0 to 0.0
Potential genotyping errors from genotypes from only 4 reads range from 0.0 to 0.0
Potential genotyping errors from genotypes from only 5 reads range from 0.0 to 0
283 number of individuals and 24830 equals 7026890 total genotypes
Total genotypes not counting missing data 6960806
Total potential error rate is between 0.0 and 0.0
SCORCHED EARTH SCENARIO
WHAT IF ALL LOW DEPTH HOMOZYGOTE GENOTYPES ARE ERRORS?????
The total SCORCHED EARTH error rate is 0.0.
Therefore, for this final dataset, we recommend the genotype quality (GQ) information provided by Stacks found in the vcf. file and processed with vcf tools to check error genotyping as shown in the figure 1. All GQ values are Phred-scaled, with a Phred value of 30 showing a genotyping error of 0.001. Therefore, genotyping error for the markers included in the dataset was basically ≤ 0.001 (with ~ 90% of the markers with an error rate ≤ 0.0001) (information not included on the text):
Figure 1. Genotype Quality (GQ) fot the 24,830 markers included in our final SNP panel
Recommendation: I believe that more clarity and discussion is needed on the treatment of admixed individuals, and the threshold of qH > 0.95 that fails to account for multigenerational hybrid individuals, no? Biologically, I think it may be wise to use HYBRIDDETECTIVE (https://doi.org/10.1111/1755-0998.12704) instead of STRUCTURE to assess the accuracy, efficiency and power to correctly identify hybrid classes/introgression. What do you think?
We are very grateful to the reviewer for this recommendation and issues raised. Indeed, the qH estimator is not designed for the determination of hybrid classes. To determine hybrid classes there are a broad range of approaches and software. One of the most popular would be NewHybrids (Anderson and Thompson 2002) with related R packages as HYBRIDDETECTIVE (Wringe et al. 2017a) and PARALLELNEWHYBRID (Wringe et al. 2017b). It is worth noting that a hybrid discrete classification would involve the assumption that only some generations of admixture have occurred (see Fitzpatrick 2012). In our scenario, this would be an unrealistic assumption since brown trout releases from hatcheries have been carried out in Iberian Peninsula for several decades with subsequent hybridisation and introgression processes previously reported (García-Marín et al. 1991; Martínez et al. 1993; Araguas et al. 2004; Araguas et al. 2017; Vera et al. 2019). Fitzpatrick (2012) described a maximum likelihood method for joint consideration of ancestry (S, the fraction of alleles derived from each parental population) together with interclass heterozygosity (HI, proportion of loci with alleles from both parental populations), using ancestry informative markers. Joint consideration of both estimates captures all the information in the discrete classification (see Table 1 in Fitzpatrick 2012), without any assumption about the number of generations of admixture. This method was implemented in HIest R package (https://cran.rstudio.com/web/packages/HIest/). We performed these analyses for our samples (N = 240 from 11 locations). Most samples (~75%) had not a robust hybrid classification but ancestry and interclass heterozygosity estimates were obtained for all samples. Nevertheless, we considered that these types of analyses would be better framed in another type of research, more focused into the biological interpretations of these results. Consequently, they were not included in the present manuscript.
### Verdict
I congratulate the authors for their outstanding work, and I appreciate their attention to detail in the analytical workflow. I believe many aquaculture genetics practitioners would like to apply the approach reported in the present study in their own study systems. It is important that you give as much detail as you can for the code used in the present study.
We thank again the very positive recommendation given by the reviewer.
References
Anderson, E.C.; Thompson, E.A. A model-based method for identifying species hybrids using multilocus genetic data. Genetics 2002, 160, 1217–1229.
Araguas, R.M.; Sanz, N.; Pla, C.; García-Marín, J.L. Breakdown of the brown trout evolutionary history due to hybridization between native and cultivated fish. J. Fish Biol. 2004, 65, 28–37.
Araguas, R.M.; Vera, M.; Aparicio, E.; Sanz, N.; Fernández-Cebrián, R.; Marchante, C.; García-Marín, J.L. Current status of the brown trout (Salmo trutta) populations within eastern Pyrenees genetic refuges. Ecol. Freshw. Fish 2017, 26, 120–132.
Fitzpatrick, B.M. Estimating ancestry and heterozygosity of hybrids using molecular markers. BMC Evol. Biol. 2012, 12, 1–14.
García-Marín, J.L.; Jorde, P.E.; Ryman, N.; Utter, F.; Pla, C. Management implications of genetic differentiation between native and hatchery populations of brown trout (Salmo trutta) in Spain. Aquaculture 1991, 95, 235–249.
Martínez, P.; Arias, J.; Castro, J.; Sánchez, L. Differential stocking incidence in brown trout (Salmo trutta) populations from Northwestern Spain. Aquaculture 1993, 114, 203–216.
Puritz, J.B.; Hollenbeck, C.M.; Gold, J.R. dDocent: a RADseq, variant-calling pipeline designed for population genomics of non-model organisms. PeerJ 2014, 2, e431.
Vera, M.; Bouza, C.; Casanova, A.; Heras, S.; Martínez, P.; García-Marín, J.L. Identification of an endemic Mediterranean brown trout mtDNA group within a highly perturbed aquatic system, the Llobregat River (NE Spain). Hydrobiologia 2019, 827, 277–291.
Wringe, B.F.; Stanley, R.E.; Jeffery, N.W.; Anderson, E.C.; Bradbury, I.R. HYBRIDDETECTIVE: A workflow and package to facilitate the detection of hybridization using genomic data in R. Mol Ecol Resour. 2017a, 17, 275–284.
Wringe, B.F.; Stanley, R.E.; Jeffery, N.W.; Anderson, E.C.; Bradbury, I.R. parallelnewhybrid: an R package for the parallelization of hybrid detection using NewHybrids. Mol Ecol Resour, 2017b, 17, 91–95.
